# Identification of heterosis and combining ability in the hybrids of male sterile and restorer sorghum [*Sorghum bicolor* (L.) Moench] lines

**Yizhong Zhang**[1,2,3], **Jing Chen**[1], **Zhenfeng Gao**[4], **Huiyan Wang**[2,3], **Du Liang**[2,3], **Qi Guo**[2,3], **Xiaojuan Zhang**[2,3], **Xinqi Fan**[2,3], **Yuxiang Wu**[1]*, **Qingshan Liu**[3]

1 College of Agronomy, Shanxi Agricultural University, Taigu, Shanxi, People's Republic of China,
2 Sorghum Research Institute, Shanxi Key Laboratory of Sorghum Genetic and Germplasm Innovation, Shanxi Agricultural University, Yuci, Shanxi, People's Republic of China, 3 National Laboratory of Minor Crops Germplasm Innovation and Molecular Breeding (in preparation), State Key Laboratory of Sustainable Dryland Agriculture, Shanxi Agricultural University, Taiyuan, Shanxi, People's Republic of China, 4 College of Food Science and Engineering, Shanxi Agricultural University, Taiyuan, Shanxi, People's Republic of China

* yuxiangwu2009@hotmail.com

**Data Availability Statement:** All relevant data are within the manuscript and its Supporting information files.

## Abstract

In sorghum [*Sorghum bicolor* (L.) Moench], combining ability and heterosis analysis are commonly used to evaluate superior parental lines and to screen for strongly heterotic hybrids, which helps in sorghum variety selection and breeding. In this context, combining ability and heterosis analysis were assessed using 14 restorer lines and seven cytoplasmic male sterile (CMS) lines in 2019 and 2020. The analysis of variance of all cross combinations had highly significant differences for all characters studied, which indicated a wide variation across the parents, lines, testers, and crosses. Combining ability analysis showed that the general combining ability (GCA) and specific combining ability (SCA) of the different parents were differed significantly among different traits. Most combinations with high SCA also showed high GCA in their parent lines. The heritability in the narrow sense of grain weight per panicle and grain yield was relatively low, indicating that the ability of these traits to be directly inherited by offspring was weak, that they were greatly affected by the environment. The better-parent heterosis for plant height, grain weight per panicle, panicle length, and 1000-grain weight was consistent with the order of mid-parent heterosis from strong to weak. The GCA effects of two lines 10480A, 3765A and three testers 0-30R, R111, and JY15R were significant for the majority of the agronomic traits including grain yield and might be used for improving the yield of grains in sorghum as parents of excellent specific combining ability. Seven strongly heterotic F$_1$ hybrids were screened; of these, hybrids 3765A × R111, 1102A × L2R, and 3765A × JY15R showed significant increases in seed iristectorigenin A content and will feature into the creation of new sorghum varieties rich in iristectorigenin A.

**Funding:** This research was supported by the Key Laboratory of Highway Construction and Maintenance Technology in the Loess Region of Shanxi Transportation Research Institute (SXBYKY2022071), the National Laboratory of Minor Crops Germplasm Innovation and Molecular Breeding (in preparation) (202204010910001-28), the National Millet Sorghum Industrial Technology System Sorghum Product Processing Post (CARS-06-14.5-A30), the Key Laboratory of Dynamic Cognitive System of Electromagnetic Spectrum Space (2023CYJSTX03-08), the Biological Breeding Project of Shanxi Agricultural University (YZGC062). The funders had no role in study design, data collection and analysis, decision to publish, or preparation of the manuscript.

**Competing interests:** The authors have declared that no competing interests exist.

## Introduction

Sorghum (*Sorghum bicolor* (L.) Moench) is the fifth most important global cereal crop after rice, wheat, maize, and barley, and it plays an important role in global food security. Sorghum is a highly adaptable crop that is widely grown in tropical, subtropical, and temperate regions [1]. As a C4 plant, sorghum has high photosynthetic efficiency and can tolerate different environmental stresses such as drought, salinity, barrenness, high temperature, and low temperature [2]. Therefore, sorghum is the crop of choice for Climate-Smart Agriculture (CSA) in the context of climate change [3]. Sorghum was one of the first crops to achieve hybrid advantage, and the successful selection of CMS lines for sorghum opened up a wide range of prospects for the application of hybrids [4]. Sorghum hybrids are widely adaptable, high yielding, and disease resistant, and play an important role in improved sorghum yields [5].

The key to hybrid breeding depends upon the identification and evaluation of superior parental lines. Parental lines can be evaluated by genetic relationship, heterosis, and combining ability [6]. The application value of any parent in hybrid breeding depends on its ability to make excellent hybrids with other parents. To effectively utilize heterosis, the systematic selection of parental lines and the identification of excellent hybrid combinations are very important [7]. The general combining ability (GCA) and special combining ability (SCA) of the parents play an important role in determining the performance of offspring [8], and are important indices by which to evaluate the utilization value of parents and the basis for combining strong heterosis in hybrids [9]. Huang et al. found that parents with higher GCA in rice can produce hybrids with higher yield, which indicates that combining ability can be used to further predict heterosis in yield traits, and can be combined with other parameters to select excellent parents in a hybrid breeding system [10]. Song et al. found that yield traits in the $F_1$ were generally biased towards parents with high GCA, and the results of a transcriptomic analysis also verified this result [11]. Liu et al. fine mapped two major GCA QTLs in rice, *GCA1* and *GCA2*, which are pseudo-response regulator genes encoding the proteins OsPRR37 and Ghd7, and found that most of the excellent rice varieties contain *GCA1* or *GCA2* [12–14]. This indicates that parents with high GCA effect can easily transfer the corresponding traits to hybrids, which explains why parents with high GCA effect are more likely to produce excellent varieties. In contrast, Azad et al. set a combining ability study in rice using 6 CMS lines × 4 testers and reported good specific cross combinations from low × low, high × low, and low × high general combine parents, respectively [15]. Similar results found from the study of Venkatesan et al. [16], which set a reported dominance and epistatic gene action controlling the characters viz., plant height, days to first flowering, grain yield per plant, panicle per plant, and grain L/B ratio.

Sorghum is a major food crop in Africa, India, and parts of China [17]. Knowledge of genetic variability [18, 19], gene action [20], heritability [21], stability [22], heterosis [15, 23], and combining ability [20] are critical for improving yield and yield contributing traits and qualitative traits like nutrients [24], proteins [25], β-carotene [26], ascorbic acids [22] phytochemicals like polyphenols [27], flavonoids [28] and antioxidants [24] in any crops breeding programs. For example, a higher magnitude of SCA than GCA variance for grain iron and zinc concentrations indicated the importance of non-additive gene action in the improvement of nutritional traits [29]. For amino acids (lysine and threonine) and mineral contents (iron and zinc), additive gene action was through to be predominant in the transmission of these traits, because GCA variance was higher than SCA variance [30]. For both stems and leaves, the GCA effect on soluble sugar was marked by strong negative correlations with GCA on cellulose, hemicellulose, and lignin [31]. In addition, sorghum has the potential to be highly nutritious and functional health components, so it is gradually emerging as a raw material for

health foods [32]. Sorghum is also a very good source of bioactive compounds that can promote human health [32]. The results of *in vitro* and animal studies have shown that compounds from sorghum, mainly phenolics, can promote beneficial changes in non-communicable disease-related parameters such as obesity, diabetes, dyslipidemia, cardiovascular disease, cancer, and hypertension [30, 33–36]. However, there is still a lack of special sorghum varieties with outstanding functional characteristics. Iristectorigenin A is a natural isoflavone that possesses good antioxidant activity and also has anti-inflammatory and anti-allergic effects [37, 38]. Although the researchers has excavated iristectorigenin A (Gao et al. unpublished) from sorghum grains through extensive targeted metabolomics in sorghum grain, the dominant parental combination that can increase the content of iristectorigenin A in sorghum grains is not clear.

Therefore, in this study, we selected 21 excellent sorghum parental lines, and 98 hybrid combinations were prepared based on line × tester mating design. The field investigation was carried out around the five main traits of the parental lines and hybrid combinations, and the combining ability effect, heterosis performance, and heritability in sorghum were analyzed. Excellent hybrid combinations and excellent parents were screened, and the content of iristectorigenin A in the different combinations was determined. The results of our study will provide a reference for sorghum hybrid breeding research.

## Materials and methods

### Experimental site

This study was conducted at Sorghum Research Institute of Shanxi Agricultural University, Yuci, Shanxi Province. The site is located at 37.58 ˚ N and 112.70˚ N in 2019 and 2020.

### Soil and climate

The study site belongs to a temperate continental monsoon climate zone and has an average annual temperature of 9.8˚C, with an annual mean precipitation of 425.0 mm that mostly occurs from July to September, and an average frost-free period of 158 days. The altitude is 803 m. The soil texture was calcareous cinnamon loam with pH = 7.6, organic matter (1.27%), total nitrogen (0.10%), and exchangeable K (0.13 cmol/kg).

### Materials

Seven CMS lines were used in crosses with fourteen restorer lines in a line × tester mating design [39] to produce 98 $F_1$ hybrids at Sanya in Hainan Province during winter 2017. During the evaluation of $F_1$ hybrids, Seven maintainer lines viz. Tx3197B, L407B, A2V4B, 1102B, 10480B, Tx623B, and 3765B along with the popular sorghum cultivar Jinza 22, were used as a check. Its combination is A2SX44A × SXR30.

Tx3197A and Tx623A are A1 cytoplasm. L407A, A2V4A, 1102A, 10480A, and 3765A are A2 cytoplasm. These two groups of parental lines were combined in different years to produce a number of excellent $F_1$ hybrid varieties for cultivation over a large area in China. The parents used carry many elite genes and have great potential for germplasm innovation. They are also representative of parental lines for the study of heterosis and combining ability in sorghum. The type and origin of the parental lines are given in Table 1.

### Synthesis $F_1$ hybrid

All the CMS parents and fourteen restorer lines were grown in the field. To avoid foreign pollen and prevent desiccation, glycine paper bags were used to cover CMS lines panicles before

**Table 1. Type and origin of the parental lines used in the test.**

| Parent | Origin | Parent | Origin | Parent | Origin |
|---|---|---|---|---|---|
| Tx3197A | USA | 5-27R | Liaoning, China | J12R | Jilin, China |
| L407A | Liaoning, China | LZ615R | Shanxi, China | J105R | Jilin, China |
| A₂V4A | Shanxi, China | SCSR | Shanxi, China | XL7R | Shanxi, China |
| 1102A | Shanxi, China | 0-30R | Liaoning, China | JL5R | Shanxi, China |
| 10480A | Shanxi, China | R111 | Shanxi, China | 1383-2R | Shanxi, China |
| Tx623A | USA | L17R | Shanxi, China | 3560R | Sichuan, China |
| 3765A | Shanxi, China | L2R | Sichuan, China | JY15R | Liaoning, China |

flowering. The next morning (9:00–11.00 a.m.), blooming panicles of respective restorer parents were collected and carried to the dusting room for pollination. At room temperature, panicles were placed in containers filled with water for 30 min to complete full blooming. Then bloomed panicles were then dusted over the emasculated panicles. Bagging and tagging were carried out in the pollinated panicles.

## Collection and preservation of $F_1$ seeds

The mature naked $F_1$ seeds were collected from the CMS parents and sun-dried. Then the seeds were oven-dried at about 28°C temperature for five days. Seeds were treated with the malathion before storage to prevent infection of store grain pests. Finally, the labeled seeds were preserved in desiccators in the cold room.

**Sowing of experimental materials seed.** In order to eliminate the impact of plant height on dwarf materials, 98 $F_1$ hybrids were randomly planted at Yuci in Shanxi Province during summer 2018, and the plant height was investigated during the maturity period and divided into two groups. Group I was for high stem materials, and Group II was for dwarf materials.

All experimental materials (98 $F_1$ hybrids divided into two groups, Group I ang Group II) were planted in a randomized complete block design with three replications in 2019 and 2020 at Yuci. The crops were sown on 10 May in 2019 and 6 May in 2020, and both were harvested during the first week of October. During this growth period, the average monthly temperature, rainfall, and sunshine duration were 20.0°C, 264.6 mm, and 1,354 h in 2019 and 19.2°C, 411.6 mm, and 1,763.8 h in 2020, respectively. The individual plot size was 4.0 m$^2$ (2 rows of 4 m length) and spacing was of 40 cm × 25 cm. A randomized block design was used with three replicates.

**Fertilizer management.** Cattle manure (45 m$^3$/ha) was applied as base fertilizer in all experimental plots, and a compound fertilizer (N-P2O5-K2O: 28-15-8; 750 kg/ha) was applied before sowing, one day before sowing, followed by rotary tillage. Urea (225 kg/ha) was applied at the jointing stage.

**Irrigation.** The irrigation method was drip irrigation, which was used four times over the entire growth period. The drip irrigation periods were after sowing, before jointing and heading, at flowering, and at the filling stage.

**Pest management.** During the occurrence period of aphids and borers, drone flight prevention was carried out by spraying 50ml of 5% acetamiprid, 50ml of 5% imidacloprid, and 150ml of 4% high chlorine emamectin benzoate per 667 m$^2$.

**Harvesting and storing of $F_1$ seeds..** In the late stage of sorghum wax ripening, when the grain moisture content is below 20%, it is harvested manually. Then threshing, cleaning, and weighing.

**Data collection.** Ten randomly selected plants from each line and hybrids were used for recording data in sorghum wax ripening period [15]. Data were recorded on plant height, panicle length, grain weight per panicle, 1000-grain weight and grain yield. The data was collected according to the "Guidelines for the conduct of tests for distinctness, uniformity and stability-Sorghum (*Sorghum bicolor (L.)* Moench)" in China.

Plant height was recorded the length from the ground to the top of the sorghum plant panicle. Panicle length was recorded the length of the stem node from the top of the panicle to the bottom of the panicle. Grain weight per panicle was recorded after harvest and air drying, single ear threshing and weighing. 1000-grain weight was recorded by harvest and air dry the ears before threshing. Randomly select 1000 seeds, accurately weigh to 0.01g, and repeat 3 times. Grain yield was recorded by randomly select ten plants for threshing and weighing.

## Qualitative analysis of iristectorigenin A for standard heterosis combination and the extraction of iristectorigenin A

The iristectorigenin A from hybrids standard heterotic combinations was extracted using an ultrasonic extraction method. In a 250 mL round-bottom blue cap flask, 10 g of dried sorghum grain that had been ground into a powder was extracted with 100 mL of ethanol for 60 min at 250 W and 50K Hz. The extracts were centrifuged at 8,500 rpm for 10 min and the EtOH phase was dried in a rotary evaporator. After settling down, the extract was dissolved in 10 mL methanol.

The iristectorigenin A content in each extract was analyzed by HPLC-DAD (diode array detector; Thermo Fisher, U3000) using an external standard method. The stationary phase was a $C_{18}$ column (150 mm × 4.6 mm, 5 μm, 100 A, Thermo Scientific Syncronis), and the mobile phases were 0.1% formic acid aqueous solution (A) and 100% methanol (B). The flow rate was 1 mL/min, and the column temperature was 25˚C. The injection volume was 100 μL and the UV detector was set at 268 nm. The gradient elution program was as follows: initial, 90% B for rinsing the column; 0–5 min, 90% B; 6–10 min, 100% B. The iristectorigenin A content of each extract was calculated with a standard curve.

A standard curve for iristectorigenin A was established using an iristectorigenin A standard (HPLC≥97%, CAS No. 39012-01-6) purchased from Shanghai Yuanye Biotechnology Co., Ltd. Standard solutions were prepared with different concentrations of iristectorigenin A (20 μg/mL, 50 μg/mL, 100 μg/mL, 200 μg/mL, 250 μg/mL, and 500 μg/mL). In this study, the standard curve for Iristectorigenin A was calculated using the formula $y = 0.00204 + 0.00109 \times x$ (S1 Fig), and the HPLC chromatograms are shown in S2 Fig.

**Statistical analyses.** Mean values of the replicates were calculated for each measurement. Analysis of variance (ANOVA) for the two years and combined ANOVA were computed using the PROC MIXED procedure in SAS 9.1.3 (SAS Institute 2003). The effects of genotype on the agronomic traits and yields were partitioned into parent and hybrid effects to assess the significance of combining ability of the parental lines. These hybrid effects were further partitioned into CMS line, restorer line, and CMS line × restorer line interaction effects, which correspond respectively to the general combining ability (GCA) effects for restorer line, the GCA effect for CMS line, and the specific combining ability (SCA) effects [39]. The estimated variance components of combining ability were analyzed using GenStat v20.1. To estimate trait heritability, the analogous broad-sense and narrow-sense coefficients of genetic determination were estimated according to Olweny et al. [40]. Significance tests for GCA and SCA effects were performed using a *t*-test. Mid-parent, better parent, and standard heterosis were calculated.

Mid-parent heterosis (MPH) was calculated as the difference between the $F_1$ hybrid mean and the average of its parents [41] as follows:

$$\text{Mid parent heterosis (MPH)} = [(F_1 - MP)/MP] \times 100$$

Where $F_1$ is the mean of the $F_1$ hybrid performance and heterosis (MP) = (parent 1 + parent 2)/2 in which parents 1 and 2 are the means for the inbred parents, respectively.

$$\text{Better parent heterosis (BPH)} = [(F_1 - HP)/HP] \times 100$$

where HP = mean of the better parent.

Standard heterosis (SH) was used to estimate genetic gain or superiority of the hybrids to standard varieties in a given area. SH was computed as:

$$SH = [(F_1 - SV)/SV] \times 100$$

Where SV = mean of the standard variety.

## Results

### Analysis of variance for the five agronomic traits

Table 2 represents the analysis (ANOVA) for the five agronomic traits. Except for replications variance of plant height and grain yield, highly significant genotypic differences were found for all the parameters. Significant mean squares of the genotypes were observed for plant height, panicle length, grain weight per panicle, 1000-grain weight, and grain yield all of which indicated the preponderance of genetic variations across the genotypes and justified the inclusion of the genotypes under study. Crosses, lines, testers, and lines × testers had highly significant differences for all the traits which specified that crosses significantly differed from each other.

### Estimate of the GCA/SCA variance ratio and heritability

The genetic parameters of five traits are shown in Table 3. The significant SCA and GCA variance was observed for all the characters studied, indicating both non-additive and additive gene action are involved in these traits. The GCA variances for plant height, panicle length, and grain yield were high, 76.27%, 86.89%, and 66.11%, respectively, indicating that the additive effect of the parents played a leading role in the inheritance of these traits. The inheritance of plant height, thousand-grain weight, and grain yield was mainly affected by the GCA effects of restorer line. The panicle length was mainly affected by the GCA effects of CMS line, and the grain weight per panicle was affected by the GCA effects of CMS line and restorer line. In comparison, the narrow heritability of plant height, panicle length and 1000-grain weight were higher, indicating that they were genetically stable and less affected by the environment, and could be selected in the early generation [42].

### Analysis of general combining ability (GCA) effects

The GCA effect of the CMS line and restorer line parents for all measured traits are presented in Table 4. Negative GCA were compulsory for plant height, although positive GCA were required for other traits included in the study. None of the CMS lines was observed to be a good general combiner (GC) parent for all the traits studied. Tx3197A exhibited negative and significant GCA effects for plant height indicated as good GC parents for shorter plant stature. CMS lines, 10480A, Tx623A, and 3765A had positive and significant GCA effects for panicle length. Among them, the panicle length of 10480A has the highest GCA. Similarly, L407A,

**Table 2. Mean squares from the combined analysis of variance (ANOVA) for the five sorghum traits.**

| Source of variance | df | Plant height (cm) | Panicle length (cm) | Grain weight per panicle (g) | 1000-grain weight (g) | Grain yield (t/ha) |
|---|---|---|---|---|---|---|
| Replications | 2 | 1.620 | 10.321** | 90.314** | 49.751** | 3.510 |
| Crosses | 97 | 19.430** | 17.742** | 7.286** | 12.509** | 2.321** |
| Lines | 6 | 127.003** | 227.874** | 53.553** | 73.69** | 5.234** |
| Testers | 13 | 135.601** | 66.87** | 24.188** | 74.186** | 3.318** |
| Lines × Testers | 78 | 14.121** | 9.563** | 6.561** | 11.249** | 4.098** |

\* and \*\*, significant at the level of $p < 0.05$ and $p < 0.01$, respectively; df, degrees of freedom.

1102A, 10480A, and 3765A displayed positive and significant GCA for grain weight per panicle, indicating that these four lines could be used as a good GC for more grain weight panicle. In the case of 1000-grain weight, only 3765A displayed positive and significant GCA. 1102A, 10480A, and 3765A displayed significant GCA effects for grain yield and these three lines could be used as good GC lines for improving the grain yield of sorghum.

No good GC testers were observed for any traits included in the study. Testers, 5-27R, SCSR, 0-30R, L17R, and J12R displayed negative and significant GCA for plant height. So, these five testers was a useful GC for short plant stature. Testers J105R and XL7R had positive and significant GCA effects for panicle length. The testers 0-30R, XL7R, JL5R, and JY15R displayed positive significant general combining effects (GCA) for grain weight per panicle. Tester R111, XL7R, and JY15R displayed positive and significant GCA effects for 1000-grain weight and might be selected as a good GC parent for large grain. Tester 0-30R, R111, L2R, 3560R, and JY15R displayed positive and significant GCA for grain yield. These testers could be used as a good GC for improving grain yield of sorghum. Considering grain yield and its contributing traits, the line 3765A and tester 0-30R were the best GC parents.

## Analysis of specific combining ability (SCA) effects

The estimates of SCA effects for the five traits in 2019 and 2020 presented in S1 Table. For each trait, evidently the plant height featured a larger SCA range, for which more hybrids had significant SCA effects ($p < 0.05$) than other traits. Negative SCA effects were required for

**Table 3. Estimates of genetic parameters for the five measured traits.**

| Traits | Genotypic variance | | | Environmental variance | Variance in combining ability of the group (%) | | Ratio $V_g/V_s$ | Heritability | |
|---|---|---|---|---|---|---|---|---|---|
| | $\sigma^2 GCA_c$ | $\sigma^2 GCA_r$ | $\sigma^2 SCA_{c \times r}$ | $V_E$ | $V_g$ (%) | $V_s$ (%) | | $H^2$ (%) | $h^2$ (%) |
| Plant Height | 92.24** | 24.79** | 36.41** | 159.89 | 76.27 | 23.73 | 3.21 | 88.97 | 57.35 |
| Panicle length | 1.27** | 2.64** | 0.59** | 5.65 | 86.89 | 13.11 | 6.63 | 84.30 | 55.80 |
| Grain weight per panicle | 39.11** | 39.91** | 39.97** | 240.41 | 56.41 | 43.59 | 1.29 | 73.11 | 41.99 |
| 1000-grain weight | 3.31** | 0.85** | 2.11** | 12.75 | 46.37 | 53.63 | 0.86 | 72.96 | 51.88 |
| Grain yield | 0.18** | 0.03** | 0.10** | 0.58 | 66.11 | 33.89 | 1.95 | 74.75 | 42.97 |

$\sigma^2 GCA_c$, GCA variance of CMS line; $\sigma^2 GCA_r$, GCA variance of restorer line; $\sigma^2 SCA_{c \times r}$, SCA variance of CMS line × restorer line; $H^2$, broad-sense heritability; $h^2$, narrow-sense heritability; $V_g$, the rate of GCA variance; $V_s$, the rate of SCA variance.

\* significant at 5% level,

\*\* significant at 1% level.

**Table 4. General combining ability (GCA) effects of parents (lines and testers) for the five measured traits.**

| Parent | Plant Height | Panicle length | Grain weight per panicle | 1000-grain weight | Grain yield |
|---|---|---|---|---|---|
| Lines | | | | | |
| Tx3197A | -18.72** | -3.93** | -23.47** | 0.07 | -0.62** |
| L407A | 2.12** | -0.94** | 6.37** | -2.18** | -0.23* |
| A2V4A | 10.56** | -1.95** | -4.82* | 0.48 | 0.05 |
| 1102A | -0.20 | -0.11 | 9.28** | 0.68 | 0.24* |
| 10480A | -0.24 | 4.85** | 5.34* | -2.62** | 0.39** |
| Tx623A | 5.10** | 0.77** | 1.03 | 1.07* | -0.08 |
| 3765A | 1.37 | 1.30** | 6.26** | 2.51** | 0.24* |
| SE | 1.544 | 0.495 | 4.465 | 1.024 | 0.210 |
| SE (gi-gj) | 2.037 | 0.623 | 5.890 | 1.350 | 0.277 |
| Testers | | | | | |
| 5-27R | -26.15** | -0.73* | -6.18 | -1.26 | -0.96** |
| LZ615R | 1.45 | 1.80** | -3.41 | -1.93** | -0.82** |
| SCSR | -15.94** | -3.52** | -20.23** | -2.97** | -0.83** |
| 0-30R | -8.70** | -0.62 | 10.02** | 1.81* | 0.42** |
| R111 | 3.85** | -0.44 | 6.77* | 4.41** | 0.53** |
| L17R | -14.29** | 0.02 | -7.27* | 1.58* | 0.02 |
| L2R | 6.59** | -2.16** | 7.39* | 0.91 | 0.88** |
| J12R | -27.48** | -1.70** | -24.53** | -7.83** | -1.11** |
| J105R | 4.87** | 4.74** | 6.48* | -1.33 | -0.11 |
| XL7R | 10.26** | 2.21** | 8.89** | 3.83** | 0.35* |
| JL5R | 17.59** | 0.20 | 10.15** | 1.88* | -0.01 |
| 1383-2R | 3.49** | 0.01 | -4.43 | -2.79** | -0.29 |
| 3560R | 11.45** | 1.29** | -0.69 | -0.41 | 0.44** |
| JY15R | 33.02** | -1.08** | 17.02** | 4.1** | 1.52** |
| SE | 2.185 | 0.700 | 6.315 | 1.450 | 0.297 |
| SE (gi-gj) | 2.882 | 0.924 | 8.330 | 1.912 | 0.392 |

* significant at 5% level,

** significant at 1% level.

plant height, while positive SCA effects were desirable for other traits included in the study. None of the hybrids was observed to be a good specific combiner (SC) for all the traits studied. The hybrid 3765A × LZ615 displayed the maximum negative and significant SCA effect for plant height and was observed as the best SC for dwarf plant height. L407A × L17R, Tx623A × J12R, Tx3197A × JY15R, 1102A × JL5R, A2V4A × L17R displayed negative and significant SCA effects and were found to be good SC hybrids for dwarf plant height. On the other hand, Tx623A × JY15R displayed the positive and maximum significant SCA effect for plant height. But when we are breeding sorghum hybrids, if the plant height is too high, it is easy to lodging. The hybrids Tx3197A × 1383-2R, 1102A × 5-27R, 10480A × L17R, Tx623A × 0-30R had the positive and greater significant SCA effects for panicle length. These four crosses could be used as the best hybrids for panicle length. The hybrids 10480A × L17R, A2V4A × L2R, 1102A × 5-27R, L407A × LZ615R had positive and significant SCA effects for grain weight per panicle and might be considered good SC hybrids. Three hybrids A2V4A × JL5R, 1102A × JY15R, Tx3197A × SCSR exhibited high positive and significant SCA effects for 1000-grain weight. These hybrids could be used as a good SC for improving the

1000-grain weight of sorghum. The hybrid 3765A×3560R exhibited the maximum positive and significant SCA effects for grain yield and were considered the best hybrids for grain yield. The best heterotic hybrid was produced from the high × high GC parents, indicating additive gene actions were involved in the cross. Seven hybrids Tx623A × XL7R, 1102A × JY15R, A2V4A × L2R, A2V4A × XL7R, 1102A × L2R, 10480A×L17R, and 1102A×0-30R exhibited high positive and significant SCA effects for grain yield and were considered good hybrids for grain yield. The hybrids 1102A×JY15R, 1102A×0-30R, 3765A×3560R produced from the crosses of high × high GC parents also. Across these good hybrids, i.e., 1102A × L2R, and 10480A × L17R were produced from the crosses of high × low GC parents. Three hybrids Tx623A × XL7R, A2V4A × L2R, A2V4A × XL7R were produced from the crosses of low ×high GC parents.

## Performance of parental lines and hybrids

The ranges for the major traits in all hybrids and parental lines are presented in Table 5, S2 and S3 Tables. The CV for grain weight per panicle was the highest (26.3%–30.8%), followed by that of thousand-grain weight (18.4%–16.9%). The plant heights for the CMS line, restorer line, and hybrids ranged from 96.0–144.0 cm, 109.3–155.7 cm, and 117.7–231.7 cm, respectively. The plant heights of hybrids made with J12R and 5-27R as restorer line were the lowest, and the heights of hybrids made with JY15R and JL5R as restorer line were the highest. Hybrids made with 3765A, 10480A, and J105R as parents had the longest panicle lengths, and hybrids made with 3765A, 10480A, JY15R, and L17R had the highest grain weight per panicle. Hybrids made with 3765A and 0-30R as parents had the highest thousand-grain weights. Comparing the traits of the parents and the hybrids showed that plant height and grain weight per panicle in the hybrids were significantly higher than in the parents, indicating that heterosis for these two traits was particularly significant.

## Heterosis analysis

Mid-parent heterosis (MPH) and better-parent heterosis (BPH) were estimated where the hybrids and parents were evaluated in adjacent experiments. Sorghum hybrids show obvious MPH and BPH and the heterosis of different traits varied greatly (Table 6 and S4 Table). The MPH of plant height was the strongest. The variation in grain weight per panicle was the largest, ranging from −43.81 to 84.68. The BPH of plant height, panicle length, grain weight per panicle and thousand-grain weight were −2.23 to 66.49, −17.61 to 30.31, −40.6 to 106.41, and −71.38 to 21.48, respectively. All hybrids showed positive MPH for plant height, and more than 50 of the hybrids were positive for MPH for grain weight per panicle, panicle length, and

**Table 5. Average performance of the parental lines and F$_1$ hybrids in yield-related traits from 2019 to 2020.**

| Traits | CMS lines | | | | Restorer lines | | | | Hybrids | | | |
|---|---|---|---|---|---|---|---|---|---|---|---|---|
| | Min | Max | Mean | CV (%) | Min | Max | Mean | CV (%) | Min | Max | Mean | CV (%) |
| Plant Height (cm) | 96.0 | 144.0 | 118.4 | 12.9 | 109.3 | 155.7 | 132.1 | 11.0 | 117.7 | 231.7 | 178.7 | 12.5 |
| Panicle length (cm) | 23.2 | 34.4 | 27.8 | 13.2 | 19.7 | 35.6 | 25.3 | 17.3 | 23.3 | 43.0 | 30.9 | 11.4 |
| Grain weight per panicle(g) | 32.5 | 76.1 | 57.44 | 30.8 | 37.0 | 84.4 | 64.3 | 26.5 | 35.7 | 137.9 | 82.8 | 26.3 |
| 1000-grain weight (g) | 23.3 | 35.1 | 28.9 | 16.9 | 21.0 | 39.2 | 30.7 | 18.4 | 20.9 | 42.9 | 29.8 | 16.1 |

The data in the table is the average value of three years from 2019 to 2020.

**Table 6. Mean, minimum, and maximum mid-parent heterosis and better-parent heterosis for agronomic traits of 98 crosses.**

| Traits | MPH | | | | | BPH | | | | |
|---|---|---|---|---|---|---|---|---|---|---|
| | Min (%) | Max (%) | Mean (%) | No. of positive heterosis | No. of negative heterosis | Min (%) | Max (%) | Mean (%) | No. of positive heterosis | No. of negative heterosis |
| Plant Height (cm) | 12.57 | 77.14 | 40.26 | 98 | 0 | −2.23 | 66.49 | 29.95 | 97 | 1 |
| Panicle length (cm) | −2.28 | 36.34 | 13.87 | 94 | 4 | −17.61 | 30.31 | 4.94 | 66 | 32 |
| Grain weight per panicle(g) | −43.81 | 84.68 | 18.67 | 78 | 20 | −40.6 | 106.41 | 9.36 | 65 | 33 |
| 1000-grain weight (g) | −67.76 | 29.62 | 0.73 | 55 | 43 | −71.38 | 21.48 | -7.57 | 28 | 70 |

thousand-grain weight. Among them, there were only 28 positive combinations for thousand-grain weight, indicating that the thousand-grain weight of hybrids was difficulty than that of high parents, and most hybrids tended to be mid-parent.

## Selection of hybrids based on standard heterosis

Through a comprehensive evaluation of plant height, panicle length, grain weight per panicle, thousand-grain weight, and grain yield, we selected 15 hybrids in which the yield was increased by >5% compared with the control (Table 7, S5 Table). There were four hybrids with 1102A as the CMS line. The yield of 1102A × JY15R was 35.6% higher than the CK, but the plant height was >200 cm, which increases the risk of lodging. The yields of 1102A × L2R, 1102A × 0-30R, and 1102A × R111 were 33.6%, 29.5%, and 7.6% higher than the CK, respectively. The plant height was around 180 cm, which is relatively good in hybrids. There were three hybrids with 3765A as the CMS line, 3765A × 3560R, 3765A × JY15R, and 3765A × R111, in which the plant height was around 180 cm, which is >5.0% higher than in the control. Compared with CK, the yields of 10480A × L17R, 10480A × JY15R, and

**Table 7. Estimates of standard heterosis of 15 crosses for grain yield than 5%.**

| Crosses | standard heterosis/% | | | | |
|---|---|---|---|---|---|
| | Plant height | Panicle length | Panicle weight | 1000-grain weight | Grain yield |
| 1102A × JY15R | 28.19 | 10.18 | 26.80 | -3.58 | 35.55 |
| 1102A × L2R | 10.81 | -1.09 | 33.63 | -6.19 | 33.64 |
| 3765A × 3560R | 20.38 | 29.09 | 48.58 | -14.01 | 29.50 |
| 1102A × 0-30R | 13.38 | 14.91 | 20.10 | -14.01 | 26.81 |
| A2V4A × L2R | 36.25 | 1.82 | 35.70 | -3.91 | 26.03 |
| 10480A × L17R | 11.69 | 25.09 | 16.49 | -3.58 | 19.69 |
| 3765A × JY15R | 35.25 | 7.64 | 27.32 | 9.77 | 18.51 |
| 10480A × JY15R | 19.88 | 11.64 | 29.25 | -14.01 | 17.67 |
| A2V4A × XL7R | 25.88 | 4.36 | 14.43 | -2.28 | 15.44 |
| 10480A × R111 | 11.38 | 26.55 | 27.19 | 0.00 | 8.59 |
| Tx623A × XL7R | 12.25 | 18.55 | -1.80 | -27.04 | 7.94 |
| 1102A × R111 | 9.63 | 14.55 | 17.78 | 10.75 | 7.57 |
| Tx623A × JY15R | 20.38 | 2.91 | 38.92 | -68.73 | 7.21 |
| Tx623A × R111 | 15.63 | 13.82 | 8.63 | -2.93 | 6.63 |
| 3765A × R111 | 17.13 | 8.73 | 10.44 | 11.73 | 6.21 |

10480A × R111 increased by 19.7%, 17.7%, and 8.6%, respectively, and they are ideal hybrids. Plant height was around 180 cm. In addition, A$_2$V4A and Tx623A were also combined in five F$_1$ hybrids, but the plants were too tall to meet the hybrid breeding target. In general, the restorer line JY15R and R111 and the CMS lines 1102A, 3765A, and 10480A had high GCA and showed great potential to make F$_1$ hybrids that display strong heterosis for yield.

## Determination of iristectorigenin A contents in standard heterosis hybrids

The results of HPLC analysis showed that the content of iristectorigenin A in the 3765A × R111 hybrid was the highest, followed by 1102A × L2R and 3765A × JY15R, and all were significantly higher than the content in the control (A2SX44A × SXR-30) ($p < 0.05$) (Fig 1 and S3 Fig). In addition, among the 15 hybrids, 1102A × JY15R and 3765A × R111 showed the opposite trend in the advantages of yield and iristectorigenin A content. Although 1102A × JY15R had the highest yield, the content of iristectorigenin A was lower, while grain yield of 3765A × R111 was relatively low, but it had the highest iristectorigenin A content. 1102A × L2R showed good advantages in both grain yield and iristectorigenin A content. The yield-increasing effect of 3765A × JY15R was moderate, but the content of iristectorigenin A was high. Therefore, considering the field traits and iristectorigenin A content, 3765A × R111, 3765A × JY15R, and 1102A × L2R are recommended as the preferred hybrids for the breeding of new sorghum varieties rich in iristectorigenin A.

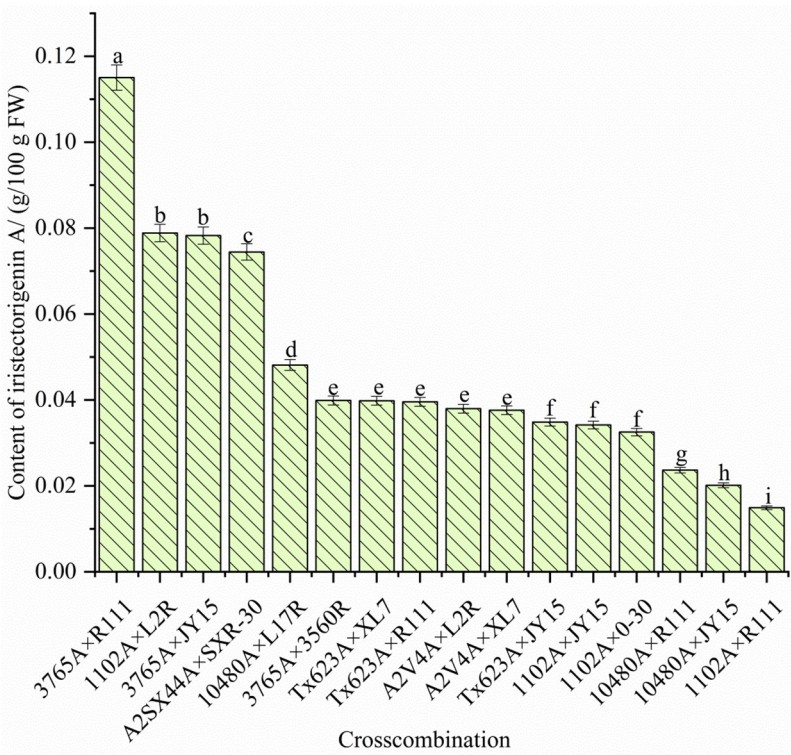

**Fig 1. Differences in iristectorigenin A contents in 15 different standard heterosis hybrids compared to the Jinza 22 (A2SX44A × SXR30).** Different lower-case letters above the error bars indicate significant differences among the treatments at $p < 0.05$.

## Discussion

Highly significant differences for the five agronomic traits indicated the existence of considerable variation in the parents and hybrids. These findings corroborated with the results of previous workers [43–45]. GCA variation is derived from the additive effect of traits with high heritability [46]. In contrast, Significant SCA effects indicated the non-additive gene action due to dominance or over dominance gene effects in the hybrids which are considered for selection of superior hybrids [47, 48]. The GCA variance of panicle length and plant height was 86.89% and 76.27%, respectively, indicating that the additive gene effect played a leading role in the inheritance of these traits which were in agreement with the results of the previous worker [49]. Both the additive and non-additive gene effects were important for GWPS and GY. On the one hand, previous studies have shown that additive gene effects are relatively more important for plant height and grain yield in sorghum [50, 51]. Erenso [52] and Medraoui [53] reported additive gene action for plant height. However, non-additive gene action was predominant in controlling plant height, grain yield, thousand-seed weight, and panicle weight [53]. On the other hand, other studies reported that both additive and non-additive gene effects were important in determining panicle length, grain yield, and 100-grain weight in sorghum [52, 54].

The results of our study are similar to those of previous studies, but there are also differences. It can be seen that the genetic mechanisms that control the major traits of sorghum are very complex and breeders in different countries utilize sorghum materials from different genetic backgrounds. The performance of quantitative traits such as GWPS and GY are susceptible to environmental and cultivation conditions, which means that the results of different studies may vary. High GCA/SCA and predictability ratios for PH and PL indicate the predominance of additive gene action and stronger transmission from parent to offspring, allowing for the election of elite lines in early generations which were in agreement with the results of the previous worker [15]. The relatively low narrow-sense heritability for grain weight and yield per panicle suggests that these traits are weakly transmitted directly to offspring and are more subject to environmental influences that can be accomplished at higher generations.

Information on the GCA effects of the parental lines helps breeders to estimate the genetic potential of breeding material for many desired traits [53]. The differences in GCA between lines are mainly due to additive genetic effects and higher order additive interactions [41]. At present, most studies of combining ability have focused on the identification of promising parental lines [55, 56]. GCA effects have widely been used by breeders to evaluate potential breeding parents [20]. The consensus of opinion is that elite breeding parents possess preferred GCA effects, which are characteristics of wide adaptability, high hereditary capacity, and elite agronomic traits [53]. Among the CMS line parents used in this study, 10480A and 3756A exhibited higher and positive GCA effects for panicle length and grain yield, but 10480A exhibited negative GCA effects for plant height. Hence, these parents can be effectively used in breeding the high yielding and medium straw sorghum hybrids, which are excellent CMS line parents, identified good general combiner parents due to high GCA for yield and quality traits of rice and maize, respectively [20, 57, 58]. The highly significant and positive GCA effects for 1000-grain weight observed in 3765A may be may be useful in developing large grain hybrids. Among the restorer line parents, JY15, R111, and L2R had highly significant and positive GCA estimates for grain yield and grain weight per panicle, and they are all excellent restorer line parents. The hybrids 3765A × 3560R had significant SCA effects for grain yield. The hybrid Tx623A × 0–30 was the highest SCA effects for panicle length. The hybrid L407A×LZ615 was the highest SCA effects for grain weight per panicle. The Tx3197A×SCSR hybrid had the highest SCA effect for 1000-grain weight. It seems not necessarily that parents with higher GCA

have a higher probability of forming high SCA hybrids [59]. Interestingly, the better SCA-effects of grain yield across eight good hybrids, three hybrids, i.e., 1102A×JY15R, 1102A×0-30R, 3765A×3560R produced from the crosses of high × high GC parents. The hybrids 1102A × L2R, and 10480A × L17R were produced from the crosses of high × low GC parents. Three hybrids Tx623A × XL7R, A2V4A × L2R, A2V4A × XL7R were produced from the crosses of low ×high GC parents, indicating both additive and non-additive gene actions were involved in these cross combinations. Especially, 10480A × L17R, the CMS line 10480A had positive and significant for panicle length, grain weight per panicle, and Grain yield, but the male parent L17R has negative GCA values for all other traits except for a higher GCA value for 1000 grain weight. These results were corroborative to the previous findings of rice [15], where they reported good specific cross combinations from low × low, high × low, and low × high general combiner parents, respectively.

A reasonable level of heterosis for grain yield and related traits is critical in any hybrid breeding program. The degree of heterosis is therefore determined by the genetic diversity present within the germplasm collection being used. Quinby (1963) reported heterosis of 39% to 80% for grain yield, and Blum et al. (1990) reported heterosis of 23.9% to 39.6% for grain yield. Blum et al. also observed significant heterosis for biomass, grain yield per plant, and grain number per panicle [60]. Wang et al. reported that MPH and BPH for grain yield, grain weight per ear, and plant height were stronger than for other traits in sweet sorghum, and that MPH and BPH for thousand-grain weight were the lowest (17.24% and 7.23%, respectively) [59]. The results of our study showed that the important agronomic traits in sorghum hybrids had obvious MPH and BPH. The MPH and BPH for plant height were the highest, at 40.26% and 29.95%, respectively, and the heterosis was positive in all cases. The second trait was grain weight per panicle, and the MPH and BPH were 18.67% and 9.36%, respectively. MPH and BPH for thousand-grain weight were the lowest at 0.73% and −7.57%, respectively. The number of hybrids in which MPH was positive and negative was basically the same. This indicates that the inheritance of thousand-grain weight in sorghum is complex; some hybrids tended to have the mid-parent value, while some tended to have the better-parent value.

None of the $F_1$ sorghum hybrids were found to have high and desirable SCA effects for all of the combined traits investigated in our study; thus, grain yield was singled out and used as an important selection criterion for hybrid performance. Grain yield is the most important target trait in most breeding programs [60]. Superior hybrids were selected based on both hybrid performance and the SCA effects of the hybrids. Through the comprehensive evaluation of plant height, panicle length, grain weight per panicle, thousand-grain weight, and grain yield, we selected 15 hybrids with yield increases of >5% by using the standard advantage of yield. Of the 15 hybrids, we found that seven had better overall performance and could be further evaluated and demonstrated. In general, improving the GCA of the parental lines is the key to sorghum germplasm innovation. On the basis of high GCA, the dominant combination of high SCA is selected, and the utilization of sorghum heterosis will make it possible to achieve greater breakthroughs.

Sorghum grain is a rich source of beneficial bioactive compounds such as phenolic acids, flavonoids, proanthocyanidins, and stilbenoids [61]. With the continuous focus on natural foods and human health, there is a growing demand for sorghum in the food shops of advanced countries around the world. The accumulating body of literature demonstrating that sorghum has advantageous features such as antioxidant, anti-inflammatory, and anti-proliferative activity and reduced glycemic index is anticipated to expand the prospects for consumption of sorghum as a more important part of the human diet [62–64]. Hence, the breeding of new varieties with a focus on functional components will be another important direction in the field of sorghum breeding.

Significant positive mid-parent heterosis for grain Fe concentration indicated that there could be an opportunity to exploit heterosis for improving grain Fe levels [29]. In determining the combining ability of hybrid lines for grain yield and its components, the male parents KO-BC1-F6-1053, SB-BC1-F6-1090, SB-BC1-F6-1036, SB-BC1-F6-1053 and the female parent 216-2AP4-5 showed good general combining ability (GCA) for the studied traits. Moreover, the identified crosses and parents are suitable for use in the development of superior hybrids, and breeding materials containing high lysine, threonine, iron, and zinc contents [30]. For both stems and leaves, the GCA effect on soluble sugar content was marked by strong negative correlations with GCA for cellulose, hemicellulose, and lignin. Positive correlations of the GCA effect were found between cellulose, hemicellulose, and lignin. The GCA effect on ash was negatively correlated with soluble sugar but positively correlated with hemicellulose in stems, whereas in leaves the GCA effect on ash had significant negative correlations with soluble sugar, cellulose, and hemicellulose [31]. Although these studies showed that the nutritional characteristics of sorghum can be significantly improved by optimizing the parental combination of $F_1$ hybrids, the factors that determine contents of iristectorigenin A in the parents and hybrids in sorghum is still unknown. Several empirical uses of iristectorigenin A have been validated through in vitro and in vivo studies, demonstrating that iristectorigenin A exhibits potent antioxidant, anticancer, hepatoprotective, neuroprotective, antidiabetic, and antimicrobial properties [65]. Therefore, in this study, we used 'Jinza 22' as the control, and the HPLC external standard method was used to identify the 15 superior hybrid combinations with excellent field traits. Three superior hybrid combinations (3765A × R111, 1102A × L2R, and 3765A × JY15R) with high iristectorigenin A contents were selected for the first time, and these three superior hybrids will have obvious advantages in the breeding of new functional varieties rich in iristectorigenin A. These three hybrids can be used as an important resource for developing lines high in functional components such as iristectorigenin A.

## Conclusions

Significant SCA and GCA variances were obtained for all traits, indicating both non-additive and additive gene action are involved in these traits. The GCA and SCA of the different sorghum lines used as parents were significantly different among the studied traits, and the MPH and BPH of the hybrids were significant. CMS lines, 10480A and 3765A displayed significant GCA for grain yield and the majority of agronomic traits can be used as good general combiner lines for improving the grain yield of sorghum. The restorer line, 0-30R, R111, and JY15R displayed significant GCA effects for grain yield and grain weight per panicle, and the majority of agronomic traits can be used as good general combiner restorer lines for improving the grain yield of sorghum. A comprehensive evaluation of important traits such as plant height, panicle length, grain weight per panicle, thousand-grain weight, and grain yield, allowed us to select seven excellent hybrids over standard check variety Jinza 22, which could be further evaluated and demonstrated. In addition, 3765A × R111, 1102A × L2R, and 3765A × JY15R had significantly increased contents of iristectorigenin A in the grains, and will feature in the breeding of new functional varieties rich in iristectorigenin A.

## Supporting information

**S1 Fig. HPLC chromatograms for the standard solutions of iristectorigenin A at six different concentrations.**
(PDF)

**S2 Fig. The standard curve for iristectorigenin A used in this study.**
(PDF)

**S3 Fig. All chromatograms of iristectorigenin A content of 15 heterotic hybrids.**
(PDF)

**S1 Table. Estimates of specific combining ability effects (SCA) for measured characters.**
(PDF)

**S2 Table. Average performance of parents related traits (2019–2020).**
(PDF)

**S3 Table. Average performance of hybrids related traits (2019–2020).**
(PDF)

**S4 Table. Estimates of mid-parent and better parent heterosis of 98 crosses for agronomic traits.**
(PDF)

**S5 Table. Estimates of standard heterosis of 98 crosses for agronomic traits.**
(PDF)

## Author Contributions

**Conceptualization:** Yizhong Zhang, Yuxiang Wu, Qingshan Liu.

**Data curation:** Yizhong Zhang, Zhenfeng Gao.

**Funding acquisition:** Yizhong Zhang.

**Investigation:** Yizhong Zhang, Jing Chen, Du Liang, Qi Guo.

**Visualization:** Yizhong Zhang, Huiyan Wang, Xiaojuan Zhang, Xinqi Fan.

**Writing – original draft:** Yizhong Zhang.

**Writing – review & editing:** Yizhong Zhang, Huiyan Wang, Yuxiang Wu.

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
