## [Decision Letter · Decision Letter 0]

18 Jul 2023

PONE-D-23-17627Identification of strong heterosis in F1 hybrids and determination of seed iristectorigenin A content of sorghum [Sorghum bicolor (L.) Moench]PLOS ONE

Dear Dr. wu,

Thank you for submitting your manuscript to PLOS ONE. After careful consideration, we feel that it has merit but does not fully meet PLOS ONE’s publication criteria as it currently stands. Therefore, we invite you to submit a revised version of the manuscript that addresses the points raised during the review process.

We look forward to receiving your revised manuscript.

Kind regards,

Mehdi Rahimi, Ph.D.

Academic Editor

PLOS ONE

Journal Requirements:

6. We note that you have referenced (unpublished) on page 5 which has currently not yet been accepted for publication. Please remove this from your References and amend this to state in the body of your manuscript: (ie “Bewick et al. [Unpublished]”) as detailed online in our guide for authors

Reviewers' comments:

Reviewer's Responses to Questions

**Comments to the Author**

1. Is the manuscript technically sound, and do the data support the conclusions?

Reviewer #1: No

2. Has the statistical analysis been performed appropriately and rigorously? 

Reviewer #1: No

3. Have the authors made all data underlying the findings in their manuscript fully available?

Reviewer #1: No

4. Is the manuscript presented in an intelligible fashion and written in standard English?

Reviewer #1: No

5. Review Comments to the Author

Reviewer #1: The authors evaluated the heterosis and combining ability in sorghum [Sorghum bicolor (L.) Moench].

I do have several suggestions that need to be addressed by major revision. Below are some specific comments to be taken into consideration by the authors:

General comments

-The authors should more emphasize on heterosis and SCA, as their main goal is to select superior hybrids.

- Introductions must be rewritten following the suggestions.

-The methodology must be rewritten in detail making several subsections.

-The authors must present adequate information in the methodology to allow replication of the estimated methods properly.

-The discussion must be improved according to the suggestions.

-There are a lot of typos and grammatical mistakes throughout the MS. These must be addressed well by an English expert.

-The authors tend to merge two or more sentences to make a complex sentence. In scientific writing, the sentence must be simple, easily understandable without any complexity.

-Add an uppercase letter “A” for all CMS lines (example L407A), and an uppercase letter “R” for all restorer lines (example LZ615R). Change the word “male” to “restorer” and “female” to “CMS line” Follow this style throughout the MS where it exists including text, tables, figures, and supplementary files. Table 1: delete the types of columns “A” and “R”.

Specific comments

Title

It must reflect all the evaluations of the experiment.

Change the title to “Identification of heterosis and combining ability in the hybrids of male sterile and restorer sorghum [Sorghum bicolor (L.) Moench] lines”

Abstract

Line 32-33: Most sorghum hybrids with high SCA have higher GCA than their parents. Meaning less sentence. Rephrase it.

Line 33: Change “The narrow heritability” to “The heritability in the narrow sense”.

Line 35-35: Delete the unnecessary statement as your objective is to identify only hybrid generation (F1 Filial generation). You don’t have any scope to test further generations.

Line 36: Change “high-parent” to “better-parent”. Follow this style throughout the MS where it exists including text, tables, figures, and supplementary files.

Line 36-38: Change “The high-parent heterosis of each trait was consistent with the order of mid-parent heterosis from strong to weak, which were plant height, grain weight per spike, spike length, and thousand-grain weight.” to “The better-parent heterosis for plant height, grain weight per spike, spike length, and thousand-grain weight was consistent with the order of mid-parent heterosis from strong to weak.”.

Line 40: Change “excellent combining ability” to “excellent specific combining ability”.

Line 41: Change “3765A×R111” to “3765A × R111”. Follow this style throughout the MS where it exists including text, tables, figures, and supplementary files.

Introduction

The introduction is too haphazard and poorly written. Re-write the introduction with the appropriate background, importance, justification, hypothesis, and objectives of the study with relevant citations reviewing the literature.

Line 47: Change “and sorghum” to “and it”.

Line 67-77: The author only mentioned that high GCA is associated with high heterotic and specific combiner hybrids. This statement is one-sided. Besides this, millions of examples of moderate to low GCA parents can produce high heterotic and specific combiner hybrids. They also must mention such literature.

Line 65: add citations to support it “determining the performance of offspring (doi.10.3390/agronomy12081797).”

Line 79-80: Add citations to support it “Knowledge of genetic variability (doi.10.3390/plants12101984, doi.10.3390/plants12112079), gene action (doi.10.3390/plants11131774), heritability (doi.10.2298/GENSR2202761K), stability (doi.10.3390/plants11182336), heterosis (doi.10.3390/agronomy12040965, R1), and combining ability (doi.10.55730/1300-011X.3044) are critical for improving yield and yield contributing traits and qualitative traits like nutrients (doi.10.3390/antiox11061206), proteins (doi.10.7717/peerj.15588), beta-carotene (doi.10.3390/antiox11061089), ascorbic acids (doi: 10.3389/fpls.2022.1016324) phytochemicals like polyphenols (doi:10.3389/fnut.2023.1057084), flavonoids (doi.10.3390/antiox12010173) and antioxidants (doi.10.3390/antiox11122434) in any crops breeding programs.

R1: M. M. Islam, U. Sarker, M. G. Rasul, M. M. Rahman. 2010. Heterosis in local boro rice (Oryza sativa L.). Bangladesh J. Pl. Breed. Genet. 23(1): 19-30

Line 103: Change “based on an incomplete diallel cross” to “based on line × tester mating design”.

Materials and methods

-Which one is the standard cultivar ‘Jinza 22’ or ‘CK (A2SX44A×SXR30)”? clarify it.

- Subsection the methodology in more parts and describe it elaborately. For instance, only current materials can be subsection into: materials, synthesis of F1 hybrids: Under this subsection make sub-subsection: seed treatments, preparation of main field and sowing of parent seeds, fertilization, irrigation, pest management, crossing to obtained F1 seeds, harvesting and storing of F1 seeds and then subsection main experiments to allow replication of the estimated methods properly.

-Describe all methodologies of statistical analyses for its reproduction.

-The authors use two lines as experimental units. How do they exclude the border effect during taking samples for minimizing the experimental errors?

-Add pest management procedures in this section.

Line 112-113: Change “a North Carolina design II (NC II) (Sprague and Tatum, 1942; Comstock and Robinson, 1948)” to “a line × tester mating design (Kempthorne, 1957)”.

Line 126: Change zero “9.8oC” to the symbol of degree “9.8 °C”. Follow this style throughout the MS where it exists including text, tables, figures, and supplementary files.

Line 135: Change “40 cm×25 cm” to “40 cm × 25 cm”. Elaborate on the abbreviation of ‘hm2’ for the first appearance.

Line 146: Add missing trait “grain yield” Elaborately write all traits with sampling methods, how do you estimate them?

Line 150: Iristectorigenin A (chemical content) is not a Quantitative trait. Yield and its related traits are quantitative traits. Change “Quantitative” to “Qualitative”. Change “over-standard” to “standard”. Follow this style throughout the MS where it exists including text, tables, figures, and supplementary files.

Line 152: Change “from over-standard heterotic combinations” to “from hybrids over standard heterotic combinations”.

Line 178: Change “male×female” to “male × female”.

Line 187: Change “(MPH)=[(F1−MP)/MP]×100” to “(MPH) = [(F1 − MP)/MP] × 100”. Follow this style throughout the MS where it exists including text, tables, figures, and supplementary files.

Line 193: Change “Over-standard heterosis (OSH)” to “standard heterosis (SH)”. Follow this style throughout the MS where it exists including text, tables, figures, and supplementary files.

Results

The results must be presented elaborately and chronologically. For instance, what does it mean when Line, tester, line × tester, and so on are significant?

Line 199: Change “p<0.01” to “p < 0.01”. Follow this style throughout the MS where it exists including text, tables, figures, and supplementary files.

Line 207-208: Delete “These abbreviations apply to the other tables below”. Each table must be self-explanatory. The authors must have to elaborate on all abbreviations in all Table captions.

Table 2: The author combined/pooled two years of data before analysis. Why there are interactions of year and other sources of variations like male, female, etc. in the Table? Change “female” to “Line”. Change “Female × male” to “Line × tester”. Follow other items in column 1 for the words “male” and “female”. Where is GCA, SCA, and GCA/SCA variance in the source of variation in the ANOVA?

Line 205: The source of variance didn’t match with the line tester analysis. Check and correct it accordingly.

Figure 1: Where is the significant test of GCA effects? Without significant tests, the results are null and void. Only significant values are important.

Line 226: Change “general combining ability (SCA) effects” to “general combining ability (GCA) effects”.

Tables 4: Change the symbol of variance “δ2” to sigma “σ2”.

Line 288: Change “Selection of over-standard heterosis hybrids” to “Selection of hybrids based on standard heterosis”. Where are the standard heterosis values? Where is Table S4? Provide it in the revised version. No need to compare the mean performance of standard variety with 21 parents for five traits. It is not the goal of this study. Delete Figures 2 and 3 from the manuscript

Table S1: Where is the significant test of SCA effects? Without significant tests, the results are null and void. Only significant values are important.

Discussion

-The authors missed several items in this section. They presented the results haphazardly. They must discuss them by explaining and interpreting the results chronologically.

-The discussion is too poor and must be improved. It should be written by interpreting the results chronologically with mechanistic explanation and citing more relevant references.

Line 332-333: Delete “used for developing the North Carolina design II (NC II)”. Follow this style throughout the MS where it exists including text, tables, figures, and supplementary files.

Line 333: Add the sentence to support your statement “These findings corroborated with the results of previous workers (doi.10.2298/GENSR2003973H, doi.10.3390/plants11212952, doi.10.2298/GENSR2001393R;).” Just after the sentence “GCA variation is derived------.”

Line 334: Add citations “heritability (doi.10.3923/pjbs.2002.1.5).

Line 334-335: Change “SCA is defined by non-additive effects, which are considered difficult to resolve in progeny due to their low heritability (Zhou et al. 2017)” to “Significant SCA effects indicated the non-additive gene action due to dominance or over dominance gene effects in the hybrids which are considered for selection of superior hybrids (R2)”.

Line 336: Change “t indicating” to “indicating”.

Line 337: Add the citation to support your statement by changing “----inheritance of these traits” to “----inheritance of these traits which were in agreement with the results of the previous worker (R3).

Line 353: Add the citation to support your statement by changing “----elite lines in early generations” to “----elite lines in early generations which were in agreement with the results of the previous worker (doi.10.3390/agronomy12040965).

Line 362: Add the citations to support your statement “GCA effects have widely been used by breeders to evaluate potential breeding parent (doi.10.3390/plants11131774).”

Line 368: Add the sentence “Sarker et al. (2002a, 2002b) and (doi.10.55730/1300-011X.3044) identified good general combiner parents due to high GCA for yield and quality traits of rice and maize, respectively.” Just before the sentence “The highly significant ----.”

Sarker, U. and M. A. K. Mian. 2002a. Line × tester analysis for yield and its components in rice (Oryza sativa L.). J. Asiat. Soc. Bangladesh Sci. 28(1): 71-81.

Sarker, U., M. G. Rasul and M. A. K. Mian. 2002b. Heterosis and combining ability in rice. Bangladesh J. Pl. Breed. Genet. 15(1): 17-26.

Line 376: Where is the discussion on mean performance, and SCA effects (the most important aspect of your study)?

R2: Alam, A. S. M. S., U. Sarker and M. A. K. Mian. 2007. Line x tester analysis in hybrid rice (Oryza sativa L.). Ann. Bangladesh Agric. 11(1): 37-44.

R3: Sarker, U., M. G. Rasul and M. A. K. Mian. 2003. Combining ability analysis of CMS and restorer lines in rice (Oryza sativa L.). Bangladesh J. Pl. Breed. Genet. 16(1): 01-07.

Conclusions

Rewrite the conclusion with major findings and conclusive remarks/recommendations of the study.

Tables: All tables must be self-explanatory. Elaborate all abbreviations in table footnotes.

References

Italicize the scientific name

In supplementary files, provide genetic parameters, mean performance, and heterosis values of all parents and hybrids separately in Tables. Also, provide all chromatograms of iristectorigenin A content of 15 heterotic hybrids.

In supplementary Table S2: provide the mean data of standard variety

6. PLOS authors have the option to publish the peer review history of their article (what does this mean?). If published, this will include your full peer review and any attached files.

Reviewer #1: No

---

## [Author Response · Author response to Decision Letter 0]

24 Nov 2023

Dear Editor and Reviewers, 

We are very grateful for your critical comments and suggestions. According to these comments and suggestions, we have made revisions on the manuscript. We accepted all the suggestions of editor and reviewers. All changes made in the manuscript are in red. Our point-by-point responses to the editor and reviewers’ comments are as follows.

Journal Requirements:

1. In your Methods section, please provide additional information regarding the permits you obtained for the work. Please ensure you have included the full name of the authority that approved the field site access and, if no permits were required, a brief statement explaining why.

Answer: The experimental site belongs to the self owned base of the Sorghum Research Institute of Shanxi Agricultural University. The experiment has been approved by the unit and no additional proof is required. 

In addition, the materials used in this experiment were collected and independently selected by our research team, and do not involve intellectual property disputes.

Answer: Thanks for your comments. We had ensured that grant numbers was correct for the awards you received for your study in the ‘Funding Information’ section.

Answer: Thanks for your comments. We do not change the data availability statement.

4. We note that you have referenced (unpublished) on page 5 which has currently not yet been accepted for publication. Please remove this from your References and amend this to state in the body of your manuscript: (ie “Bewick et al. [Unpublished]”) as detailed online in our guide for authors

http://journals.plos.org/plosone/s/submission-guidelines#loc- reference-style 

Answer: Thanks for your comments. We had changed according to the suggestion. See the section “Introduction” at line 104 of page 5.

Answer: Thanks for your comments. We had increased Supporting Information files at the end of the manuscript. See the line 531-549 of page 28.

6. I can't find it ‘Financial Disclosure’ section in the system. So, I declare here that

Financial Disclosure:

This research was supported by the Key Laboratory of Highway Construction and Maintenance Technology in the Loess Region of Shanxi Transportation Research Institute (SXBYKY2022071), the National Laboratory of Minor Crops Germplasm Innovation and Molecular Breeding (in preparation) (202204010910001-28), the National Millet Sorghum Industrial Technology System Sorghum Product Processing Post (CARS-06-14.5-A30), the Key Laboratory of Dynamic Cognitive System of Electromagnetic Spectrum Space (2023CYJSTX03-08), the Biological Breeding Project of Shanxi Agricultural University (YZGC062). The funders had no role in study design, data collection and analysis, decision to publish, or preparation of the manuscript.

Competing interests: The authors have declared that no competing interests exist.

I confirm the above information.

Supporting information

S1 Figure. HPLC chromatograms for the standard solutions of iristectorigenin A at six different concentrations.

(DOCX)

S2 Figure. The standard curve for iristectorigenin A used in this study.

(DOCX)

S3 Figure. All chromatograms of iristectorigenin A content of 15 heterotic hybrids

(DOCX)

S1 Table. Estimates of specific combining ability effects (SCA) for measured characters.

(XLSX)

S2 Table. Average performance of parents related traits (2019-2020).

(XLSX)

S3 Table. Average performance of hybrids related traits (2019-2020).

(XLSX)

S4 Table. Estimates of mid-parent and better parent heterosis of 98 crosses for agronomic traits.

(XLSX)

S5 Table. Estimates of standard heterosis of 98 crosses for agronomic traits.

(XLSX)

Reviewer 1: 

General comments

1- The authors should more emphasize on heterosis and SCA, as their main goal is to select superior hybrids.

Answer: For heterosis and SCA, we had reanalyzed and rewritten. See the section “Analysis of specific combining ability (SCA) effects” at line 275-304 of page 14-16.

2- Introductions must be rewritten following the suggestions.

Answer: The introduction had been rewritten according to the suggestion. See the section “Introduction” at line 45-118 of page 3-6.

3-The methodology must be rewritten in detail making several subsections.

Answer: The methodology had been rewritten according to the suggestion. It maked several subsections. See the section “Materials and methods” at line 119-236 of page 7-12.

4-The authors must present adequate information in the methodology to allow replication of the estimated methods properly.

Answer: In the “Materials and methods” section, the synthesis F1 hybrid, collection and preservation of F1 seeds, sowing of experimental materials seed, Data collection,

field management were detailed in several parts. 

See the section “Materials and methods” at line 119-236 of page 7-12.

5-The discussion must be improved according to the suggestions.

Answer: The discussion had been rewritten according to the suggestion. See the section “Discussion” at line 386-514 of page 21-27.

6-There are a lot of typos and grammatical mistakes throughout the MS. These must be addressed well by an English expert.

Answer: Language had been modified according to the suggestion.

7-The authors tend to merge two or more sentences to make a complex sentence. In scientific writing, the sentence must be simple, easily understandable without any complexity.

Answer: Language had been modified according to the suggestion.

8-Add an uppercase letter “A” for all CMS lines (example L407A), and an uppercase letter “R” for all restorer lines (example LZ615R). Change the word “male” to “restorer” and “female” to “CMS line” Follow this style throughout the MS where it exists including text, tables, figures, and supplementary files. Table 1: delete the types of columns “A” and “R”.

Answer: The manuscript has been revised according to the suggestion. 

Table 1 had deleted the types of columns “A” and “R” at line 139 of page 7.

Specific comments

Title

1- It must reflect all the evaluations of the experiment.

Change the title to “Identification of heterosis and combining ability in the hybrids of male sterile and restorer sorghum [Sorghum bicolor (L.) Moench] lines”

Answer: The title had revised according to the suggestion. See the title at line 1-3 of page 1.

Identification of heterosis and combining ability in the hybrids of male sterile and restorer sorghum [Sorghum bicolor (L.) Moench] lines

Abstract

2- Line 32-33: Most sorghum hybrids with high SCA have higher GCA than their parents. Meaning less sentence. Rephrase it.

Answer: We had rephrased according to the suggestion. See the sentence at line 34-35 of page 2.

Most combinations with high SCA also showed high GCA in their parent lines.

3- Line 33: Change “The narrow heritability” to “The heritability in the narrow sense”.

Answer: We had changed according to the suggestion. See the sentence at line 35 of page 2.

The heritability in the narrow sense of grain weight per panicle and grain yield was relatively low,

4- Line 35-35: Delete the unnecessary statement as your objective is to identify only hybrid generation (F1 Filial generation). You don’t have any scope to test further generations.

Answer: We had deleted according to the suggestion. See the sentence at line 36-37 of page 2.

The heritability in the narrow sense of grain weight per panicle and grain yield was relatively low, indicating that the ability of these traits to be directly inherited by offspring was weak, that they were greatly affected by the environment.

5- Line 36: Change “high-parent” to “better-parent”. Follow this style throughout the MS where it exists including text, tables, figures, and supplementary files.

Answer: We had changed according to the suggestion. See the sentence at line 38 of page 2.

We has modified according to this style throughout the manuscript where it exists including text, tables, figures, and supplementary files.

6- Line 36-38: Change “The high-parent heterosis of each trait was consistent with the order of mid-parent heterosis from strong to weak, which were plant height, grain weight per spike, spike length, and thousand-grain weight.” to “The better-parent heterosis for plant height, grain weight per spike, spike length, and thousand-grain weight was consistent with the order of mid-parent heterosis from strong to weak.”.

Answer: We had changed according to the suggestion. See the sentence at line 37-40 of page 2.

The better-parent heterosis for plant height, grain weight per panicle, panicle length, and 1000-grain weight was consistent with the order of mid-parent heterosis from strong to weak.

7- Line 40: Change “excellent combining ability” to “excellent specific combining ability”.

Answer: We had changed according to the suggestion. See the sentence at line 42 of page 3.

The GCA effects of two lines 10480A, 3765A and three testers 0-30R, R111, and JY15R were significant for the majority of the agronomic traits including grain yield and might be used for improving the yield of grains in sorghum as parents of excellent specific combining ability.

8- Line 41: Change “3765A×R111” to “3765A × R111”. Follow this style throughout the MS where it exists including text, tables, figures, and supplementary files.

Answer: We had changed according to the suggestion. See the sentence at line 43-45 of page 3.

We has modified according to this style throughout the manuscript where it exists including text, tables, figures, and supplementary files.

Introduction

9- The introduction is too haphazard and poorly written. Re-write the introduction with the appropriate background, importance, justification, hypothesis, and objectives of the study with relevant citations reviewing the literature.

Answer: We have rewritten the introduction. See the section “Introduction” at line 47-120 of page 3-6.

10- Line 47: Change “and sorghum” to “and it”.

Answer: We had changed according to the suggestion. See the sentence at line 49 of page 3.

Line 67-77: The author only mentioned that high GCA is associated with high heterotic and specific combiner hybrids. This statement is one-sided. Besides this, millions of examples of moderate to low GCA parents can produce high heterotic and specific combiner hybrids. They also must mention such literature.

Answer: We had increased according to the suggestion. See the sentence at line 79-84 of page 4.

In contrast，Azad et al. (2022) set a combining ability study in rice using 6 CMS lines × 4 testers and reported good specific cross combinations from low × low, high × low, and low × high general combine parents, respectively. Similar results found from the study of Venkatesan et al. (2007)set a reported dominance and epistatic gene action controlling the characters viz., plant height, days to first flowering, grain yield per plant, panicle per plant, and grain L/B ratio.

11- Line 65: add citations to support it “determining the performance of offspring (doi.10.3390/agronomy12081797).”

Answer: We had added citation according to the suggestion. See the sentence at line 67 of page 4.

Line 79-80: Add citations to support it “Knowledge of genetic variability (doi.10.3390/plants12101984, doi.10.3390/plants12112079), gene action (doi.10.3390/plants11131774), heritability (doi.10.2298/GENSR2202761K), stability (doi.10.3390/plants11182336), heterosis (doi.10.3390/agronomy12040965, R1), and combining ability (doi.10.55730/1300-011X.3044) are critical for improving yield and yield contributing traits and qualitative traits like nutrients (doi.10.3390/antiox11061206), proteins (doi.10.7717/peerj.15588), beta-carotene (doi.10.3390/antiox11061089), ascorbic acids (doi: 10.3389/fpls.2022.1016324) phytochemicals like polyphenols (doi:10.3389/fnut.2023.1057084), flavonoids (doi.10.3390/antiox12010173) and antioxidants (doi.10.3390/antiox11122434) in any crops breeding programs.

R1: M. M. Islam, U. Sarker, M. G. Rasul, M. M. Rahman. 2010. Heterosis in local boro rice (Oryza sativa L.). Bangladesh J. Pl. Breed. Genet. 23(1): 19-30

Answer: We had added citations according to the suggestion. See the sentence at line 86-92 of page 5.

Knowledge of genetic variability(Azam et al., 2023; Hossain et al., 2023), gene action(Azam et al., 2022), heritability (Kulsum et al., 2022), stability (Hasan et al., 2022), heterosis (Azad et al., 2022; Islam et al., 2010), and combining ability (Azam et al., 2022) are critical for improving yield and yield contributing traits and qualitative traits like nutrients (Sarker et al., 2022), proteins (Mannan et al., 2023), β-carotene (Sarker et al., 2022), ascorbic acids (Hassan et al., 2022) phytochemicals like polyphenols (Tarafder et al., 2023), flavonoids (Sarker et al., 2023) and antioxidants (Sarker et al., 2022) in any crops breeding programs.

12- Line 103: Change “based on an incomplete diallel cross” to “based on line × tester mating design”.

Answer: We had changed according to the suggestion. See the sentence at line 114-115 of page 6.

Therefore, in this study, we selected 21 excellent sorghum parental lines, and 98 hybrid combinations were prepared based on line × tester mating design.

Materials and methods

13 -Which one is the standard cultivar ‘Jinza 22’ or ‘CK (A2SX44A×SXR30)”? clarify it.

Answer: ‘Jinza 22’ is CK as the control variety for the hybrid, its combination is A2SX44A × SXR30. Manuscript had already provided detailed explanations.

See the sentence at line 135-137 of page 7.

During the evaluation of F1 hybrids, Seven maintainer lines viz. Tx3197B, L407B, A2V4B, 1102B, 10480B, Tx623B, and 3765B along with the popular sorghum cultivar Jinza 22, were used as a check. Its combination is A2SX44A × SXR30.

14- Subsection the methodology in more parts and describe it elaborately. For instance, only current materials can be subsection into: materials, synthesis of F1 hybrids: Under this subsection make sub-subsection: seed treatments, preparation of main field and sowing of parent seeds, fertilization, irrigation, pest management, crossing to obtained F1 seeds, harvesting and storing of F1 seeds and then subsection main experiments to allow replication of the estimated methods properly.

Answer: We have rewritten the methodology. See the section “Materials and methods at line 121-194 of page 6-10.

For example:

Experimental Site, Soil and Climate, Materials, Synthesis F1 Hybrid, Collection and Preservation of F1 Seeds, Sowing of experimental materials seed ,and so on.

15-Describe all methodologies of statistical analyses for its reproduction.

Answer: We have described the methodologies of statistical analyses. See the sentence at line 227-230 of page 11.

To estimate trait heritability, the analogous broad-sense and narrow-sense coefficients of genetic determination were estimated according to Olweny et al. (2017). Significance tests for GCA and SCA effects were performed using a t-test. Mid-parent, better parent, and standard heterosis were calculated.

16-The authors use two lines as experimental units. How do they exclude the border effect during taking samples for minimizing the experimental errors?

Answer: In order to minimize the impact of plant height on dwarf materials, 98 F1 hybrids were randomly planted at Yuci in Shanxi Province during summer 2018, and the plant height was investigated during the maturity period and divided into two groups. Group I was for high stem materials, and Group II was for dwarf materials. 

See the section “Sowing of experimental materials seed” at line 159-163 of page 8.

In order to eliminate the impact of plant height on dwarf materials, 98 F1 hybrids were randomly planted at Yuci in Shanxi Province during summer 2018, and the plant height was investigated during the maturity period and divided into two groups. Group I was for high stem materials, and Group II was for dwarf materials.

17-Add pest management procedures in this section.

Answer: We had added “Pest management” section. See the sentence at line 180-183 of page 9.

Pest management

During the occurrence period of aphids and borers, drone flight prevention was carried out by spraying 50ml of 5% acetamiprid, 50ml of 5% imidacloprid, and 150ml of 4% high chlorine emamectin benzoate per 667 m2.

18- Line 112-113: Change “a North Carolina design II (NC II) (Sprague and Tatum, 1942; Comstock and Robinson, 1948)” to “a line × tester mating design (Kempthorne, 1957)”.

Answer: We had changed according to the suggestion. See the sentence at line 133-135 of page 7.

Seven CMS lines were used in crosses with fourteen restorer lines in a line × tester mating design (Kempthorne, 1957) to produce 98 F1 hybrids at Sanya in Hainan Province during winter 2017.

19- Line 126: Change zero “9.8oC” to the symbol of degree “9.8 °C”. Follow this style throughout the MS where it exists including text, tables, figures, and supplementary files.

Answer: We had changed according to the suggestion. See the sentence at line 168 of page 9.

We has modified according to this style throughout the manuscript where it exists including text, tables, figures, and supplementary files.

20- Line 135: Change “40 cm×25 cm” to “40 cm × 25 cm”. Elaborate on the abbreviation of ‘hm2’ for the first appearance.

Answer: We had changed according to the suggestion. See the sentence at line 169-171 of page 9.

The individual plot size was 4.0 m2 (2 rows of 4 m length) and spacing was of 40 cm × 25 cm. A randomized block design was used with three replicates. 

We had changed 'hm2' to 'ha'. See the sentence at line 173-175 of page 9.

Cattle manure (45 m3/ha) was applied as base fertilizer in all experimental plots, and a compound fertilizer (N-P2O5-K2O: 28-15-8; 750 kg/ha) was applied before sowing, one day before sowing, followed by rotary tillage. Urea (225 kg/ha) was applied at the jointing stage.

21- Line 146: Add missing trait “grain yield” Elaborately write all traits with sampling methods, how do you estimate them?

Answer: We had added grain yield sampling methods. See the sentence at line 198 of page 10. 

Also, we had added all traits with sampling methods. See the sentence at line 193-198 of page 10. 

Plant height was recorded the length from the ground to the top of the sorghum plant panicle. Panicle length was recorded the length of the stem node from the top of the panicle to the bottom of the panicle. Grain weight per panicle was recorded after harvest and air drying, single ear threshing and weighing. 1000-grain weight was recorded by harvest and air dry the ears before threshing. Randomly select 1000 seeds, accurately weigh to 0.01g, and repeat 3 times. Grain yield was recorded by randomly select ten plants for threshing and weighing.

22- Line 150: Iristectorigenin A (chemical content) is not a Quantitative trait. Yield and its related traits are quantitative traits. Change “Quantitative” to “Qualitative”. Change “over-standard” to “standard”. Follow this style throughout the MS where it exists including text, tables, figures, and supplementary files.

Answer: We had changed according to the suggestion. See the sentence at line 199 of page 10.

Qualitative analysis of iristectorigenin A for standard heterosis combination and the extraction of iristectorigenin A.

We had changed “over-standard” to “standard”. See the sentence at line 242 of page 12.

We has modified according to this style throughout the manuscript where it exists including text, tables, figures, and supplementary files.

23- Line 152: Change “from over-standard heterotic combinations” to “from hybrids over standard heterotic combinations”.

Answer: We had changed “from over-standard heterotic combinations” to “from hybrids over standard heterotic combinations”.

See the sentence at line 201 of page 10.

The iristectorigenin A from hybrids standard heterotic combinations was extracted using an ultrasonic extraction method.

24- Line 178: Change “male×female” to “male × female”.

Answer: We had changed “male×female” to “CMS line × restorer line”. See the sentence at line 227 of page 11.

25- Line 187: Change “(MPH)=[(F1−MP)/MP]×100” to “(MPH) = [(F1 − MP)/MP] × 100”. Follow this style throughout the MS where it exists including text, tables, figures, and supplementary files.

Answer: We had changed according to the suggestion. See the sentence at line 237, 240, 244 of page 12.

We has modified according to this style throughout the manuscript where it exists including text, tables, figures, and supplementary files.

26- Line 193: Change “Over-standard heterosis (OSH)” to “standard heterosis (SH)”. Follow this style throughout the MS where it exists including text, tables, figures, and supplementary files.

Answer: We had changed according to the suggestion. See the sentence at line 242-244 of page 12.

We has modified according to this style throughout the manuscript where it exists including text, tables, figures, and supplementary files.

Results

27- The results must be presented elaborately and chronologically. For instance, what does it mean when Line, tester, line × tester, and so on are significant?

Answer: We had modified according to the suggestion. See the sentence at line 246-272 of page 12-14.

28- Line 199: Change “p<0.01” to “p < 0.01”. Follow this style throughout the MS where it exists including text, tables, figures, and supplementary files.

Answer: We had Change “p<0.01” to “p < 0.01”. See the sentence at line 257 of page 13.

We has modified according to this style throughout the manuscript where it exists including text, tables, figures, and supplementary files.

29- Line 207-208: Delete “These abbreviations apply to the other tables below”. Each table must be self-explanatory. The authors must have to elaborate on all abbreviations in all Table captions.

Answer: We had “These abbreviations apply to the other tables below”.

See the sentence at Table 1 257 of page 13.

We has modified according to this style on all abbreviations in all Table captions.

30- Table 2: The author combined/pooled two years of data before analysis. Why there are interactions of year and other sources of variations like male, female, etc. in the Table? Change “female” to “Line”. Change “Female × male” to “Line × tester”. Follow other items in column 1 for the words “male” and “female”. Where is GCA, SCA, and GCA/SCA variance in the source of variation in the ANOVA?

Answer: We had reanalysis Mean squares from the combined analysis of variance (ANOVA) for the five sorghum traits. We had changed “female” to “Line”. Change “Female × male” to “Line × tester”. See the sentence at Table 2 of page 13.

Table 3 had showed GCA, SCA, and GCA/SCA variance in the source of variation in the ANOVA. See the sentence at Table 3 269 of page 14.

31- Line 205: The source of variance didn’t match with the line tester analysis. Check and correct it accordingly.

Answer: We had reanalysis Mean squares from the combined analysis of variance (ANOVA) for the five sorghum traits. See the sentence at Table 2 of page 13.

32- Figure 1: Where is the significant test of GCA effects? Without significant tests, the results are null and void. Only significant values are important.

Answer: We had reanalysis the significant test of GCA effects. See the sentence at Table 3 of page 14-15. Significant tests had marked.

33- Line 226: Change “general combining ability (SCA) effects” to “general combining ability (GCA) effects”.

Answer: We had changed “general combining ability (SCA) effects” to “general combining ability (GCA) effects”.

See the sentence at 258-269 of page 14-15.

34- Tables 4: Change the symbol of variance “δ2” to sigma “σ2”.

Answer: We had changed the symbol of variance “δ2” to sigma “σ2”.

See the sentence at Table 3 line 270-273 of page 14.

Table 3. Estimates of genetic parameters for the five measured traits.

Traits Genotypic variance Environmental variance Variance in combining ability of the group (%) Ratio Vg/Vs Heritability

 σ2GCAc σ2GCAr σ2SCAc × r VE Vg (%) Vs (%) H2 (%) h2 (%)

Plant Height 92.24** 24.79** 36.41** 159.89 76.27 23.73 3.21 88.97 57.35

Panicle length 1.27** 2.64** 0.59** 5.65 86.89 13.11 6.63 84.30 55.80

Grain weight per panicle 39.11** 39.91** 39.97** 240.41 56.41 43.59 1.29 73.11 41.99

1000-grain weight 3.31** 0.85** 2.11** 12.75 46.37 53.63 0.86 72.96 51.88

Grain yield 0.18** 0.03** 0.10** 0.58 66.11 33.89 1.95 74.75 42.97

σ2GCAc, GCA variance of CMS line; σ2GCAr, GCA variance of restorer line; σ2SCAc × r , SCA variance of CMS line × restorer line; H2, broad-sense heritability; h2, narrow-sense heritability; Vg, the rate of GCA variance; Vs, the rate of SCA variance. * significant at 5% level, ** significant at 1% level.

35- Line 288: Change “Selection of over-standard heterosis hybrids” to “Selection of hybrids based on standard heterosis”. Where are the standard heterosis values? Where is Table S4? Provide it in the revised version. No need to compare the mean performance of standard variety with 21 parents for five traits. It is not the goal of this study. Delete Figures 2 and 3 from the manuscript

Answer: We had changed “Selection of over-standard heterosis hybrids” to “Selection of hybrids based on standard heterosis”. See the sentence at line 360 of page 19.

The standard heterosis values had added to Table S2. 

Table S4 had provided it in the revised version.

We had deleted Figures 2 and 3 from the manuscript.

36- Table S1: Where is the significant test of SCA effects? Without significant tests, the results are null and void. Only significant values are important.

Answer: We had reanalysis the significant test of SCA effects in Table S1. 

Discussion

37-The authors missed several items in this section. They presented the results haphazardly. They must discuss them by explaining and interpreting the results chronologically.

Answer: The discussion had been rewritten according to the suggestion. See the section “Discussion” at line 386-514 of page 21-27.

38- The discussion is too poor and must be improved. It should be written by interpreting the results chronologically with mechanistic explanation and citing more relevant references.

Answer: The discussion had been rewritten according to the suggestion. See the section “Discussion” at line 386-514 of page 21-27.

39- Line 332-333: Delete “used for developing the North Carolina design II (NC II)”. Follow this style throughout the MS where it exists including text, tables, figures, and supplementary files.

Answer: We had deleted “used for developing the North Carolina design II (NC II)”.

We has modified according to this style throughout the manuscript where it exists including text, tables, figures, and supplementary files.

40- Line 333: Add the sentence to support your statement “These findings corroborated with the results of previous workers (doi.10.2298/GENSR2003973H, doi.10.3390/plants11212952, doi.10.2298/GENSR2001393R;).” Just after the sentence “GCA variation is derived------.”

Answer: We had added the citations to support the statement. See the sentence at line 398-399 of page 22.

These findings corroborated with the results of previous workers (Hasan et al. 2020, Faysal et al. 2022, Rashad et al. 2020). 

41-Line 334: Add citations “heritability (doi.10.3923/pjbs.2002.1.5).

Answer: We had added the citations. See the sentence at line 400 of page 22.

GCA variation is derived from the additive effect of traits with high heritability(Haghighi et al. 2020).

42- Line 334-335: Change “SCA is defined by non-additive effects, which are considered difficult to resolve in progeny due to their low heritability (Zhou et al. 2017)” to “Significant SCA effects indicated the non-additive gene action due to dominance or over dominance gene effects in the hybrids which are considered for selection of superior hybrids (R2)”.

R2: Alam, A. S. M. S., U. Sarker and M. A. K. Mian. 2007. Line x tester analysis in hybrid rice (Oryza sativa L.). Ann. Bangladesh Agric. 11(1): 37-44.

Answer: We had changed according to the suggestion. See the sentence at line 400 of page 22.

In contrast, Significant SCA effects indicated the non-additive gene action due to dominance or over dominance gene effects in the hybrids which are considered for selection of superior hybrids (Alam et al. 2007)

43- Line 336: Change “t indicating” to “indicating”.

Answer: We had changed according to the suggestion. See the sentence at line 404 of page 22.

The GCA variance of panicle length and plant height was 86.89% and 76.27%, respectively, indicating that the additive gene effect played a leading role in the inheritance of these traits which were in agreement with the results of the previous worker (Sarker et al. 2003).

44-Line 337: Add the citation to support your statement by changing “----inheritance of these traits” to “----inheritance of these traits which were in agreement with the results of the previous worker (R3).

R3: Sarker, U., M. G. Rasul and M. A. K. Mian. 2003. Combining ability analysis of CMS and restorer lines in rice (Oryza sativa L.). Bangladesh J. Pl. Breed. Genet. 16(1): 01-07.

Answer: We had changed according to the suggestion. See the sentence at line 405-406 of page 22.

inheritance of these traits which were in agreement with the results of the previous worker (Sarker et al. 2003).

45- Line 353: Add the citation to support your statement by changing “----elite lines in early generations” to “----elite lines in early generations which were in agreement with the results of the previous worker (doi.10.3390/agronomy12040965).

Answer: We had changed according to the suggestion. See the sentence at line 422-423 of page 23.

elite lines in early generations which were in agreement with the results of the previous worker (Azad et al., 2022 ).

46- Line 362: Add the citations to support your statement “GCA effects have widely been used by breeders to evaluate potential breeding parent (doi.10.3390/plants11131774).”

Answer: We had changed according to the suggestion. See the sentence at line 432-433 of page 23.

GCA effects have widely been used by breeders to evaluate potential breeding parents (Azam et al. 2022).

47- Line 368: Add the sentence “Sarker et al. (2002a, 2002b) and (doi.10.55730/1300-011X.3044) identified good general combiner parents due to high GCA for yield and quality traits of rice and maize, respectively.” Just before the sentence “The highly significant ----.”

Sarker, U. and M. A. K. Mian. 2002a. Line × tester analysis for yield and its components in rice (Oryza sativa L.). J. Asiat. Soc. Bangladesh Sci. 28(1): 71-81.

Sarker, U., M. G. Rasul and M. A. K. Mian. 2002b. Heterosis and combining ability in rice. Bangladesh J. Pl. Breed. Genet. 15(1): 17-26.

Answer: We had changed according to the suggestion. See the sentence at line 439-441 of page 24.

Sarker et al. (2002a, 2002b) and Azam et al. (2022) identified good general combiner parents due to high GCA for yield and quality traits of rice and maize, respectively.

48- Line 376: Where is the discussion on mean performance, and SCA effects (the most important aspect of your study)?

Answer: We had rewritten according to the suggestion. Discussion on mean performance, and SCA effects had seen the sentence at line 448-460 of page 24-25.

It seems not necessarily that parents with higher GCA have a higher probability of forming high SCA hybrids (Wang et al. 2020). Interestingly, the better SCA-effects of grain yield across eight good hybrids, three hybrids, i.e., 1102A×JY15R, 1102A×0-30R, 3765A×3560R produced from the crosses of high × high GC parents. The hybrids 1102A × L2R, and 10480A × L17R were produced from the crosses of high × low GC parents. Three hybrids Tx623A × XL7R, A2V4A × L2R, A2V4A × XL7R were produced from the crosses of low ×high GC parents, indicating both additive and non-additive gene actions were involved in these cross combinations. Especially, 10480A × L17R, the CMS line 10480A had positive and significant for panicle length, grain weight per panicle, and Grain yield, but the male parent L17R has negative GCA values for all other traits except for a higher GCA value for 1000 grain weight. These results were corroborative to the previous findings of rice (Azad et al. 2022), where they reported good specific cross combinations from low × low, high × low, and low × high general combiner parents, respectively. 

Conclusions

49- Rewrite the conclusion with major findings and conclusive remarks/recommendations of the study.

Answer: We had rewritten according to the suggestion. See the sentence at line 525-540 of page 28.

Significant SCA and GCA variances were obtained for all traits, indicating both non-additive and additive gene action are involved in these traits. The GCA and SCA of the different sorghum lines used as parents were significantly different among the studied traits, and the MPH and BPH of the hybrids were significant. CMS lines, 10480A and 3765A displayed significant GCA for grain yield and the majority of agronomic traits can be used as good general combiner lines for improving the grain yield of sorghum. The restorer line, 0-30R, R111, and JY15R displayed significant GCA effects for grain yield and grain weight per panicle, and the majority of agronomic traits can be used as good general combiner restorer lines for improving the grain yield of sorghum. A comprehensive evaluation of important traits such as plant height, panicle length, grain weight per panicle, thousand-grain weight, and grain yield, allowed us to select seven excellent hybrids over standard check variety Jinza 22, which could be further evaluated and demonstrated. In addition, 3765A × R111, 1102A × L2R, and 3765A × JY15R had significantly increased contents of iristectorigenin A in the grains, and will feature in the breeding of new functional varieties rich in iristectorigenin A.

50- Tables: All tables must be self-explanatory. Elaborate all abbreviations in table footnotes.

Answer: We had rewritten according to the suggestion. 

References

51- Italicize the scientific name

Answer: We had rewritten according to the suggestion.

52- In supplementary files, provide genetic parameters, mean performance, and heterosis values of all parents and hybrids separately in Tables. Also, provide all chromatograms of iristectorigenin A content of 15 heterotic hybrids.

Answer: We had rewritten according to the suggestion. Table S2 and Table S3 had provided genetic parameters, mean performance, and heterosis values of all parents and hybrids separately.

Figure S3 provided all chromatograms of iristectorigenin A content of 15 heterotic hybrids.

53- In supplementary Table S2: provide the mean data of standard variety

Answer: We had provide the mean data of standard variety in Table S3.

Dear Editor

We are glad to hear from you and have read the issues very carefully. The comments are very helpful in improving the quality and significance of the manuscript. Now I response the editor’s comment with a point by point and in blue the reply.

1. We note that the grant information you provided in the ‘Funding Information’ and ‘Financial Disclosure’ sections do not match. Please see below for your reference.

Answer: We have ensured that our provided the updated Financial Disclosure Statement in the Author comments box or in the Cover Letter. Please update.

2. It appears that your ORCiD iD has not been validated in your Editorial Manager account and we are unable to proceed until that step is complete.

Answer: The ORCiD iD has been modified according to the prompts, as shown in the following figure.

3. Please ensure that you refer to Table 6 in your text as, if accepted, production will need this reference to link the reader to the Table.

Answer: We had ensured our refer to Table 6 in the manuscript. See at line 342 and 351 of page 18-19.

4. Please include a copy of Table 4.

Answer: We had modified to Table 4 in the manuscript. See at line 290 of page 15.

---

## [Decision Letter · Decision Letter 1]

12 Dec 2023

Identification of heterosis and combining ability in the hybrids of male sterile and restorer sorghum [ Sorghum bicolor  (L.) Moench] lines

PONE-D-23-17627R1

Dear Dr. wu,

We’re pleased to inform you that your manuscript has been judged scientifically suitable for publication and will be formally accepted for publication once it meets all outstanding technical requirements.

Kind regards,

Mehdi Rahimi, Ph.D.

Academic Editor

PLOS ONE

Additional Editor Comments (optional):

Reviewers' comments:

Reviewer's Responses to Questions

**Comments to the Author**

1. If the authors have adequately addressed your comments raised in a previous round of review and you feel that this manuscript is now acceptable for publication, you may indicate that here to bypass the “Comments to the Author” section, enter your conflict of interest statement in the “Confidential to Editor” section, and submit your "Accept" recommendation.

Reviewer #1: All comments have been addressed

2. Is the manuscript technically sound, and do the data support the conclusions?

Reviewer #1: Yes

3. Has the statistical analysis been performed appropriately and rigorously? 

Reviewer #1: Yes

4. Have the authors made all data underlying the findings in their manuscript fully available?

Reviewer #1: Yes

5. Is the manuscript presented in an intelligible fashion and written in standard English?

Reviewer #1: Yes

6. Review Comments to the Author

Reviewer #1: The authors revised the manuscript following the suggestion made on it. It is now substantially improved. It can be published in its current form. However, some typos such as spacing between number and unit ("0.01g" must be changed to "0.01 g", 28), a space before third bracket, change "50ml" to "50 mL", a space before degree symbol. scientist name "L." never be italicized must be addressed during proofreading.

7. PLOS authors have the option to publish the peer review history of their article (what does this mean?). If published, this will include your full peer review and any attached files.

Reviewer #1: No

---

## [Editor Report · Acceptance letter]

22 Dec 2023

PONE-D-23-17627R1 

PLOS ONE

Dear Dr. wu, 

I'm pleased to inform you that your manuscript has been deemed suitable for publication in PLOS ONE. Congratulations! Your manuscript is now being handed over to our production team.

Kind regards, 

on behalf of

Associate Prof. Mehdi Rahimi 

Academic Editor

PLOS ONE